# Nonlinearity synergy: An elegant strategy for realizing high-sensitivity and wide-linear-range pressure sensing

Rui Chen [1], Tao Luo [1], Jincheng Wang [1], Renpeng Wang [1], Chen Zhang [1], Yu Xie[1], Lifeng Qin [1], Haimin Yao [2] ✉ & Wei Zhou [1] ✉

Flexible pressure sensors are indispensable components in various applications such as intelligent robots and wearable devices, whereas developing flexible pressure sensors with both high sensitivity and wide linear range remains a great challenge. Here, we present an elegant strategy to address this challenge by taking advantage of a pyramidal carbon foam array as the sensing layer and an elastomer spacer as the stiffness regulator, realizing an unprecedentedly high sensitivity of 24.6 kPa$^{-1}$ and an ultra-wide linear range of 1.4 MPa together. Such a wide range of linearity is attributed to the synergy between the nonlinear piezoresistivity of the sensing layer and the nonlinear elasticity of the stiffness regulator. The great application potential of our sensor in robotic manipulation, healthcare monitoring, and human-machine interface is demonstrated. Our design strategy can be extended to the other types of flexible sensors calling for both high sensitivity and wide-range linearity, facilitating the development of high-performance flexible pressure sensors for intelligent robotics and wearable devices.

The keen tactile sensing of humans relies on numerous mechanoreceptors that can detect pressure in a wide range (up to 300 kPa) with a low detection limit (down to 1 Pa)[1–4]. In robotics, the achievement of dexterity of robots demands high-performance flexible sensors for pressure sensing[5–7]. For example, dexterous robotic manipulation such as grasping objects with unforeseen weight or fragility necessitates pressure sensors with a high sensitivity and a wide linear range[8]. The past five years witnessed the fast development of flexible pressure sensors from low sensitivity and narrow range to high sensitivity and wide range[9–14]. However, pressure sensors with high sensitivity normally exhibit a narrow range or poor linearity[15–22], while those with a wide linear range tend to have low sensitivity[23–31].

To date, various methods have been developed to enhance the performance of the flexible pressure sensor. To enhance the sensitivity of flexible pressure sensors, a variety of surface topological microstructures, such as pyramidal[32–34], interlocked[35,36], cylindrical[37], and domed microstructures[38], were employed in the pressure sensing layer. However, sensors based on these surficial microstructures normally exhibit sensitivity lower than 10 kPa$^{-1}$, which is the minimum sensitivity requirement for tactile sensing in dexterous robotic manipulation[8]. For higher sensitivity, sensing layers with interior microscopic porous structures and therefore notable deformability were adopted in pressure sensors[39–41]. Nevertheless, the aforementioned strategies, rooted in either surficial topological microstructures or interior microscopic porosity, suffer from a narrow linear range no more than 100 kPa. To extend the linear range, hybrid surficial topographical microstructures, which combine micro-dome and micro-cone arrays, were applied in the sensing layers, resulting in a remarkable extension of the linear range to 1 MPa. However, sensors based on the hybrid surficial topographical microstructures show a sensitivity of only around 0.3 kPa$^{-1}$ [23]. To reconcile the intrinsic conflict between the high sensitivity and wide linear range in the traditional flexible pressure sensors, sensing layers with a hierarchical microstructure based on a porous lattice structure were applied and proven to be an

[1]Pen-Tung Sah Institute of Micro-Nano Science and Technology, Xiamen University, Xiamen 361102, China. [2]Department of Mechanical Engineering, The Hong Kong Polytechnic University, Hung Hom, Kowloon, Hong Kong SAR, China. ✉e-mail: mmhyao@polyu.edu.hk; weizhou@xmu.edu.cn

effective strategy for achieving a moderate sensitivity of 4.7 kPa$^{-1}$ across a broad linear range of 1 MPa[31]. For a further augment of sensitivity within this ultra-wide linear range of 1 MPa, a hybrid hierarchical structure integrating microscopic gradient pores and pyramidal surficial microstructure was employed within a flexible sensor[42], yielding a sensitivity surpassing 10 kPa$^{-1}$. However, fabricating the gradient pores with high controllability in its geometry was proven challenging, which thereby compromised the reproducibility and reliability of this strategy. In addition, the principle governing the efficacy of this strategy remain obscure, making it quite difficult to further improve and optimize this approach.

Herein, we conceive a novel strategy for developing flexible piezoresistive pressure sensors with both high sensitivity (>10 kPa$^{-1}$) and wide linear range (>1 MPa). First, to achieve high sensitivity, we adopted a double-sided pyramidal carbon foam (DPyCF) array as the sensing layer, which integrates tapering microstructure[43-48] and microscopic porosity[49-53], two prevalent approaches to enhancing sensitivity of the piezoresistive materials. The hierarchal 3D porous structure and high compressibility of such carbon foam-based sensing layer endow it with a highly nonlinear piezoresistivity[48,54,55]. On the other hand, to extend the range of pressure sensing, we introduced an elastic spacer surrounding the sensing layer, called stiffness regulator (SR), to regulate the load share on the sensing layer. The requirement for high sensitivity at low pressure as well as the pressure sensing at high pressure necessitates a stiffening behavior (namely nonlinear elasticity) of the SR under compression. The synergy between the nonlinearities in the elasticity of the SR and the piezoresistivity of the sensing layer gives rise to the high linearity of the piezoresistive pressure sensor.

To demonstrate the aforementioned strategy, we developed a prototype following this strategy, which exhibits a sensitivity as high as 24.6 kPa$^{-1}$ and excellent linearity [coefficient of determination ($R^2$) > 0.99] in a range from 0 to 1.4 MPa. Our sensor was demonstrated capable of grasping and lifting not only rigid metal blocks (~900 g) but also soft and fragile tofu (~40 g). Moreover, our sensor was found able to measure various physiological pressure signals ranging from ~200 Pa to ~1.2 MPa. A $4 \times 4$ array of our sensors was applied to a keypad of a password lock, realizing a code-pressure double encryption. The design philosophy behind our sensor for achieving both high sensitivity and wide linear range can be further extended to the piezocapacitive pressure sensors and beyond.

## Results

### Accomplishment of high sensitivity together with a wide linear range

Figure 1a shows the structural design of our sensor (see Supplementary Fig. 1 for the details of the DPyCF@SR sensor fabrication process), which consists of a double-sided pyramidal carbon foam (DPyCF) array serving as the sensing layer and an elastomeric (Ecoflex rubber) spacer serving as the stiffness regulator (SR). The sensing layer was fabricated using a 3D dynamic focusing laser technology followed by a pyrolysis process, with ultrastructure and composition being characterized by scanning electron microscopy (SEM), energy dispersive spectrometer, Raman spectroscopy, and X-ray diffractometer (Supplementary Figs. 2 and 3). The sensing layer is nested inside the SR (Supplementary Fig. 4) and together are packaged by a 50 μm thick polyimide (PI) film with a pair of 20/30 nm Cr/Au electrodes, resulting in a pressure sensor called DPyCF@SR. The piezo-resistivity of the sensing layer was characterized (Supplementary Fig. 5), and the electrical resistance ($R$) shows an ultra-nonlinear decay with the applied compressive strain ($\varepsilon$), which can be perfectly ($R^2$ > 0.98) fitted by an exponential function as:

$$R = R_0 \exp(-\varepsilon/\alpha) \tag{1}$$

where $R_0$ is the resistance at zero strain and $\alpha$ is the decay constant characterizing the decaying rate of the resistance with the compressive

strain. It was demonstrated that the decay constant ($\alpha$) can be controllably regulated in a range from 0.12 to 0.22 by tuning the aspect ratio (height over base) of the micro-pyramid (Supplementary Fig. 6 and Supplementary Note 1). The relative change of electrical current ($\triangle I/I_0$) caused by compressive strain ($\varepsilon$), according to Ohm's law, is thereby given by (Fig. 1b):

$$\frac{\triangle I}{I_0} = \frac{R_0 - R}{R} = \exp(\varepsilon/\alpha) - 1 \tag{2}$$

On the other hand, the mechanical behavior of the ensemble of SR and the sensing layer (DPyCF) under compression was also characterized (Supplementary Fig. 10). The relationship between the nominal pressure ($p$), which is defined as the applied force divided by the area enclosed by the outer perimeter of the SR, and the compressive strain ($\varepsilon$) exhibits a nonlinear dependence, which can be perfectly ($R^2$ > 0.99) described by an exponential function as (Fig. 1b):

$$p = \beta E_0 \left[\exp(\varepsilon/\beta) - 1\right] \tag{3}$$

where $E_0$ is the tangential modulus of the ensemble of SR and the sensing layer (DPyCF) at zero strain and $\beta$ is the stiffening constant. It was demonstrated that the stiffening constant ($\beta$) can be controllably regulated in the range from 0.11 to 0.18 by tuning the mixing ratio of two building compositions when preparing the SR (Supplementary Fig. 11). Based on Eqs. (2) and (3), the sensitivity ($S$) of the sensor then is given by

$$S \equiv \frac{d(\triangle I/I_0)}{dp} = \frac{1}{\alpha E_0} \exp\left[\left(\frac{1}{\alpha} - \frac{1}{\beta}\right)\varepsilon\right] \tag{4}$$

Equation (4) indicates that the dependence of the sensitivity ($S$) on the compressive strain ($\varepsilon$) vanishes when $\alpha = \beta$, resulting in the ideal linearity ($R^2 = 1$). Guided by this theoretical finding, we designed a DPyCF@SR pressure sensor by using a sensing layer and SR with decay constant $\alpha = 0.117$ and stiffening constant $\beta = 0.122$ (Supplementary Fig. 12a, b). The as-prepared sensor exhibits a linear variation ($R^2 = 0.999$) of $\triangle I/I_0$ in an ultra-wide range from 0 to 1.4 MPa with a high sensitivity of 24.6 kPa$^{-1}$, as shown in Fig. 1c. The linear sensing factor (LSF), the product of the sensitivity and the linear sensing range[56], of our DPyCF@SR sensor, reaches 34440, which overweighs the values of any pressure sensors found in the literature[9-21,23-31,57], as shown in Fig. 1d and Supplementary Table 1.

Equation (4) also implies that the linearity of the sensor degrades with the discrepancy between $\alpha$ and $\beta$. We numerically calculated the linearity of the DPyCF@SR sensor with different combinations of $\alpha$ and $\beta$ and found that a satisfactory linearity (e.g., $R^2$ > 0.99) can still be ensured as long as $\beta$ falls in a range as follows (Fig. 1e):

$$0.68\alpha + 0.01 < \beta < 2.7\alpha^{1.5} + 0.04 \tag{5}$$

The above condition was experimentally verified by six DPyCF@SR sensors with different combinations of decay constant ($\alpha$) and stiffening constant ($\beta$) (Supplementary Fig. 13 and Supplementary Table 2).

### Performance of pressure sensing

To demonstrate the respective importance of the DPyCF sensing layer and the SR in securing the high sensitivity and wide linear range of the pressure sensor, we conducted a comparative study with three control samples. We first replaced the sensing layer with a plain carbon foam and kept the SR unchanged, the resulting pressure sensor (denoted as CF@SR) shows a much lower sensitivity and a similar linear range as compared to the DPyCF@SR sensor (Fig. 2a). We then removed the SR from the sensor and applied the DPyCF as the sensing layer, the

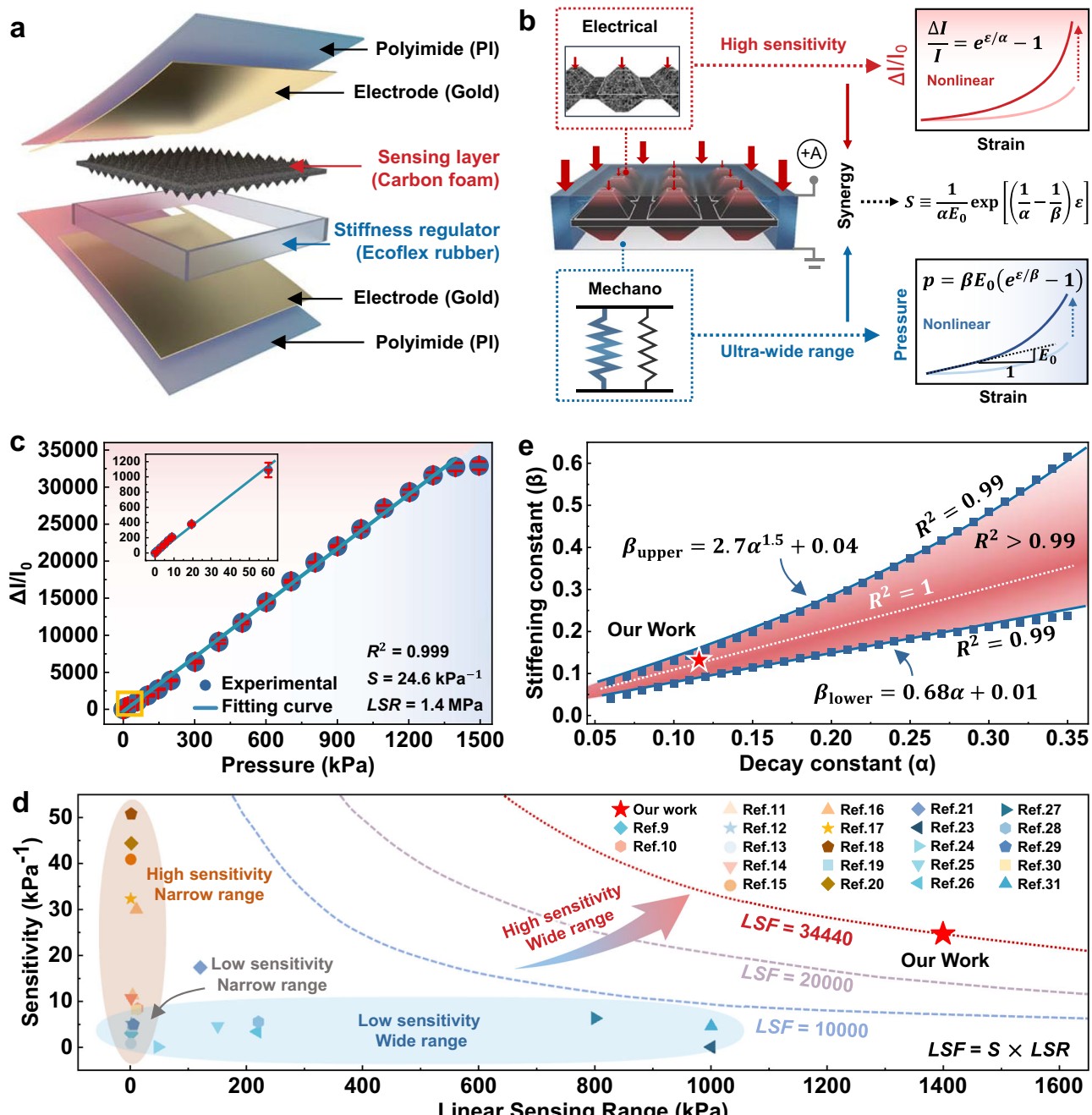

**Fig. 1 | Design and the sensing mechanism of the DPyCF@SR pressure sensor.**
**a** Exploded view illustrating the design layout of the DPyCF@SR sensor.
**b** Schematics showing the strategy for achieving high sensitivity and wide linear range together. The synergy between the nonlinear elasticity of the SR and the nonlinear piezoresistivity of the DPyCF@SR results in the high linearity of the sensor. **c** Relative current change as a function of pressure ranging from 0 to 1400 kPa as measured from one DPyCF@SR sensor. S: sensitivity, LSR: linear sensing range (n = 3 samples; center, mean; error bars, s.d.). **d** Comparison of the performance between our DPyCF@SR pressure sensor and those reported in the literatures. LSF: linear sensing factor. **e** Numerically calculated linearity of the DPyCF@SR sensors with different combinations of decay constant ($\alpha$) and stiffening constant ($\beta$).

resulting SR-free pressure sensor (denoted as DPyCF) exhibits a higher sensitivity but a much narrower linear range as compared to the DPyCF@SR sensor (Fig. 2a). Moreover, we adopted an SR with a larger stiffening constant ($\beta = 0.21$) and the same DPyCF sensing layer with decay constant ($\alpha = 0.11$), the resulting pressure sensor (denoted as DPyCF@SR2) shows a lower sensitivity and poor linearity as compared to the DPyCF@SR sensor (Fig. 2a). Above results demonstrate that the synergy of the nonlinearities in the piezo-resistivity of the sensing layer and elasticity of the SR is indispensable for ensuring high sensitivity and wide linear range together.

To evaluate the stability and robustness of the DPyCF@SR sensor, a cyclic pressure with an amplitude of 130 kPa and frequency of 2 Hz was applied to the sensor. The output signal shows that even after 50,000 loading/unloading cycles, the sensor still works well without any notable degradation in performance (Fig. 2b). To further examine the anti-fatigue performance of our sensor, we performed a cyclic compression test with an elevated pressure of 1 MPa. The results (Supplementary Fig. 14) show that the DPyCF@SR sensor can still output an undamped electrical signal after ~7800 cycles. Structural characterization showed that the integrity of the pyramidal porous structure of the DPyCF

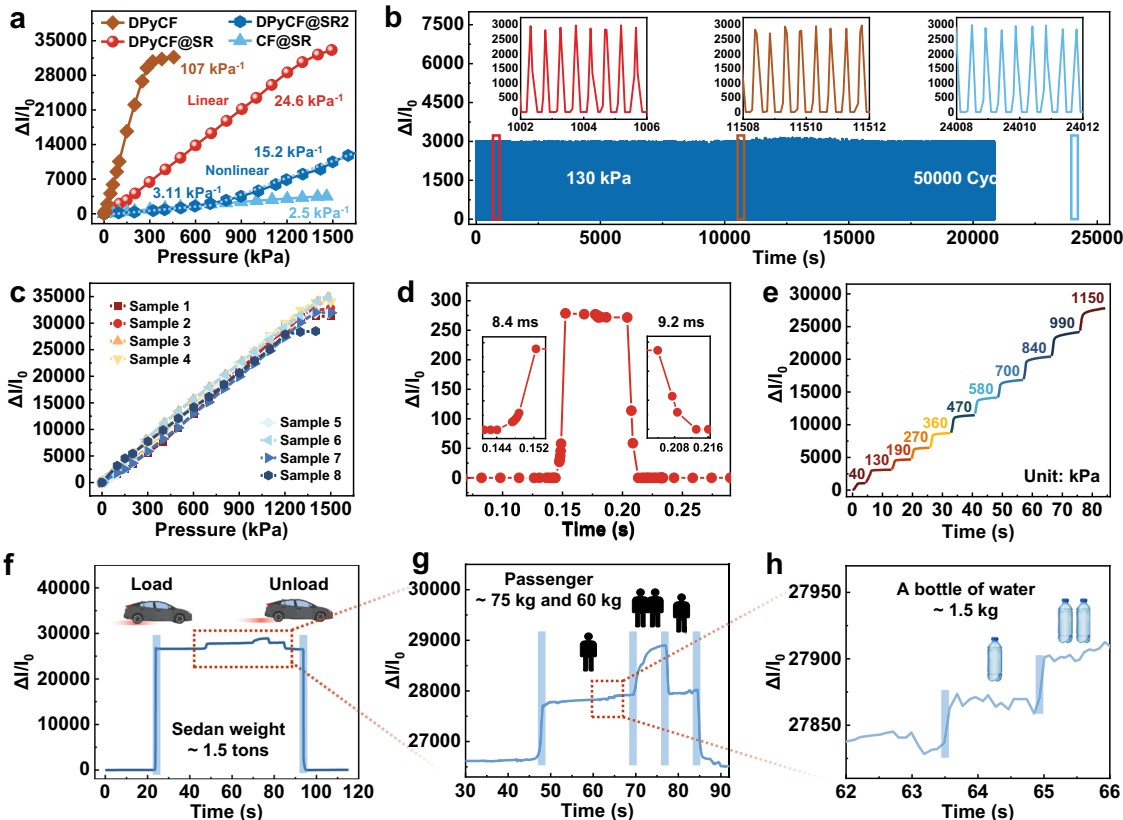

**Fig. 2 | Sensing performance of the DPyCF@SR pressure sensor. a** The relative current change as a function of pressure of the piezoresistive sensors with different structural designs: DPyCF@SR, CF@SR, DPyCF, and DPyCF@SR2. **b** Stability of the DPyCF@SR sensor under a cyclic pressure with an amplitude of 130 kPa and frequency of 2 Hz over 50,000 cycles. **c** The consistent performance of eight DPyCF@SR sensors reflects the reproducibility of the high sensitivity and wide linear range. **d** Dynamic response of the DPyCF@SR sensor. **e** A stepwise pressure increasing from 40 kPa to 1,150 kPa was detected by the DPyCF@SR sensor. **f–h** Variation of the relative current that reflects the changes of the load weight in a 1.5-ton sedan (Supplementary Movie 1).

sensing layer was well preserved during the fatigue test (Supplementary Fig. 15). The excellent reproducibility of the high sensitivity and ultra-wide linearity of the DPyCF@SR sensor was demonstrated by the consistent performance of eight samples we prepared, as shown in Fig. 2c. To evaluate the dynamic response of the DPyCF@SR, we gently placed a mass block (~70 g) on a DPyCF@SR sensor followed by a quick lift. The response and recovery times of the sensor are 8.4 ms and 9.2 ms respectively (Fig. 2d), which were faster than those of the human skin (30–50 ms)[58]. With its high sensitivity and wide linear range, our DPyCF@SR sensor can measure a stepwise pressure increment from 40 kPa to 1150 kPa and the output was considerably stable (Fig. 2e). In addition, the repeatability and accuracy of the DPyCF@SR sensor were verified at diverse pressure levels of 4, 56, 225, and 670 kPa (Supplementary Fig. 16). The pressure detection limit was measured to be 1 Pa even in a cyclic loading mode (Supplementary Fig. 17), which is comparable to a commercial high-resolution pressure sensor (SMT 200N-S, AiLogics, USA). More importantly, our DPyCF@SR pressure sensor was found able to detect a very tiny pressure even at a highly pressurized state. This unique performance of the DPyCF@SR allowed us to successfully detect a small change (~1.5 kg) of the load weight in a 1.5-ton sedan (Supplementary Fig. 18, Fig. 2f–h, and Supplementary Movie 1). Such a high sensitivity (0.09%) in sensing a pressure change in a large pressure not only exceeds the scope of most traditional pressure sensors but also surpasses the sensitivity of human skin (7%)[59].

## Tactile sensing for robotic manipulation

Dexterous robotic manipulation requires high-sensitivity and wide-range detection of the interactive force between the robot and objects, which is commonly realized by employing multiple pressure sensors. This greatly increases the hardware complexity and the manufacturing cost of the robot. The high sensitivity and wide linear range of DPyCF@SR sensor enable us to achieve dexterous robotic manipulation with one sensor.

To demonstrate the versatility of our DPyCF@SR sensor in sensing both tiny force and large force for multi-tasking adaptation of robotic manipulation, we designed a robotic grasping system, in which a robotic gripper equipped with a DPyCF@SR sensor was applied to grasp a tofu (bean curd) block (~40 g) and a steel block (~900 g), two objects with distinct features, as shown in Fig. 3a. For comparison, we adopted a commercial pressure sensor (Tekscan 5027, USA) as a control sample. We designed a closed-loop algorithm (Supplementary Fig. 19) to control the grasping force of the gripper based on the pressure sensing signal received by the pressure sensors (Fig. 3b). The process begins with a lower initial grasping force, followed by a sensor-guided lifting trial. If the lifting is unsuccessful, a higher grasping force will be applied, followed by another lifting trial. Such a grasping-and-lifting process will be repeated until a pressure signal with little fluctuation is detected (see Supplementary Note 2 for details closed-loop algorithm).

Figure 3c shows the pressure detected by the DPyCF@SR sensor during the grasping-and-lifting process of the tofu block. The curve shows a stable pressure signal which indicates that the DPyCF@SR sensor accurately perceives the forces at different stages in the grasping process, enabling the gripper to grasp the tofu block successfully (Supplementary Movie 2). In contrast, the pressure signal detected by the Tekscan sensor is quite unstable (Fig. 3d), which fails

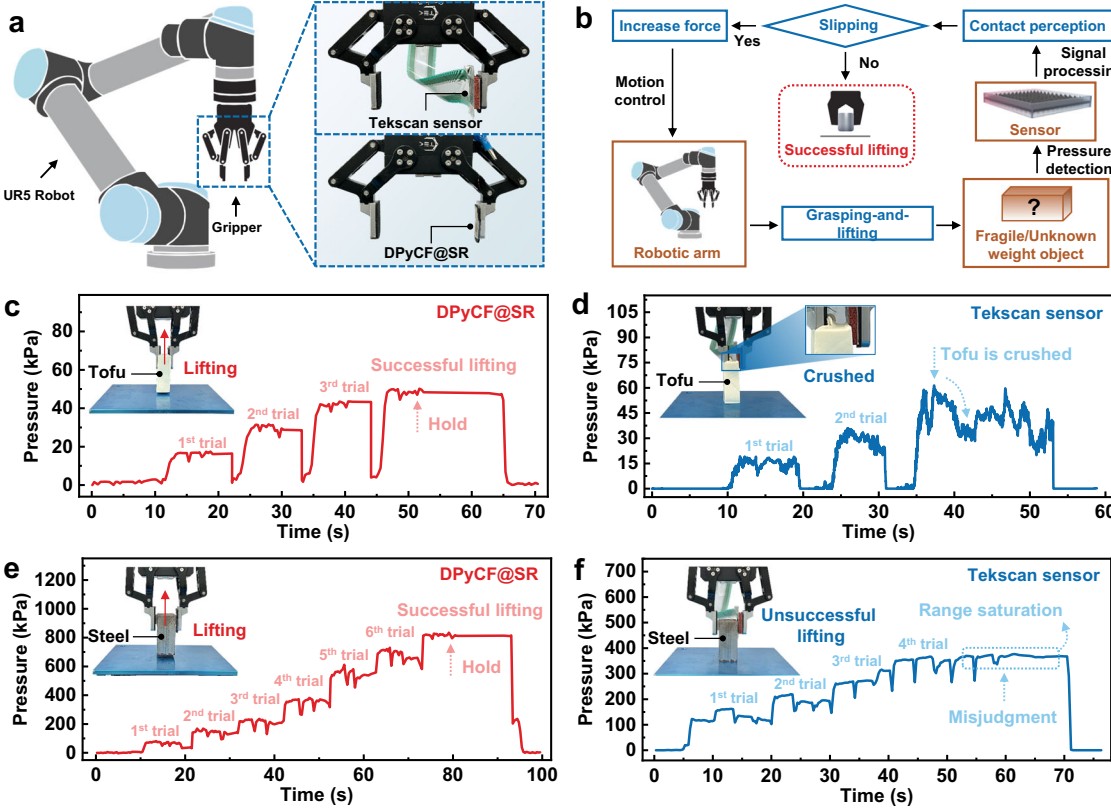

**Fig. 3 | Robotic grasping of objects. a** Schematic diagram of UR5 robot grasping system equipped with DPyCF@SR sensor or Tekscan sensor. **b** Schematic showing the robotic grasping closed-loop control scheme. **c** Robotic gripper equipped with a DPyCF@SR sensor successfully grasps and lifts a soft tofu block (-40 g). **d** Robotic gripper equipped with a Teksacn sensor fails to grasp and lift a soft tofu block (-40 g), resulting in the crush of the tofu block. **e** Robotic gripper equipped with a DPyCF@SR sensor successfully grasps and lifts a steel block (-900 g). **f** Robotic gripper equipped with a Teksacn sensor fails to grasp and lift the steel block (-900 g) due to the saturation of measurement range.

to guide the gripper to exert an appropriate grasping force on the tofu block, resulting in an unsuccessful lifting and crush of the tofu block (Supplementary Movie 3).

Figure 3e displays the pressure recorded by DPyCF@SR sensor in the grasping-and-lifting process of the steel block (-900 g). The DPyCF@SR sensor with its ultra-wide linear range accurately detects the gradually increasing grasping pressure (100–800 kPa), dictating the gripper to exert sufficient gripping force on the steel block (Supplementary Movie 4). In contrast, the Teksacn sensor shows an unchanging pressure peak (-375 kPa) from 40 s, implying the saturation of its measurement range (375 kPa). Based on such a false pressure input, the control algorithm would make a misjudgment that the gripper has grasped the object. The procedure thus is terminated, resulting in an unsuccessful lifting (Fig. 3f and Supplementary Movie 5).

### Physiological signal monitoring

The high sensitivity and wide linear range of our DPyCF@SR pressure sensor make it quite suitable for monitoring the physiological pressure signals of humans which range from a few hundred pascal to megapascal (Fig. 4a).

The pulse waveforms of the radial artery are often used in non-invasive and real-time diagnosis of cardiovascular problems, such as hypertension and arteriosclerosis[60,61]. The high sensitivity and fast response of our DPyCF@SR pressure sensor enable the capture of key features of the pulse waveform. Figure 4b shows the real-time current response (-200 Pa) of a DPyCF@SR sensor attached to the wrist of a volunteer. It can be seen that the pulse signals with an average frequency of -78–80 beats per minute can be precisely captured. Moreover, the typical pulse waveforms, including the percussion wave (P

wave), tidal wave (T wave), and diastolic wave (D wave), can be successfully recognized (inset of Fig. 4b), implying the great potential of the DPyCF@SR sensor in healthcare monitoring and disease diagnosis. Figure 4c shows the pressure signal detected by a DPyCF@SR sensor attached to the inner side of a mask worn by a volunteer. The current signals synchronizes with the respiratory pace, indicating the potential of our sensor to monitor the respiratory status. We also attached the DPyCF@SR sensor to the knee joint of a volunteer to monitor knee flexion. The repetitive triangle wave signals with flexion-dependent amplitude were obtained (Fig. 4d), implying the high flexibility of the sensor and its capability for quantitative detection of body motion. Moreover, we evaluated the detection capability of the DPyCF@SR sensor under higher pressure (-1.2 MPa) by attaching a sensor to the sole of a volunteer. A repetitive square wave with stable amplitude was received when the volunteer was walking (Fig. 4e and Supplementary Movie 6), indicating the great application potential of the DPyCF@SR sensor in pedometer and tread monitoring

### Code-pressure double encryption

Most keypad locks can be readily unlocked by an unauthorized person who knows the code numbers, implying a significant security risk. This issue could be addressed by introducing the keystroke pressure as a biometric authentication so as to realize the code-pressure double encryption.

To demonstrate this concept, we developed a keypad (Fig. 5a), which consists of 4 × 4 DPyCF@SR sensor array (see Supplementary Fig. 20 for details on the sensor array fabrication process). All the sensors exhibit a consistent current change in response to the applied pressure (Fig. 5b). A microcontroller was used for data collection, and

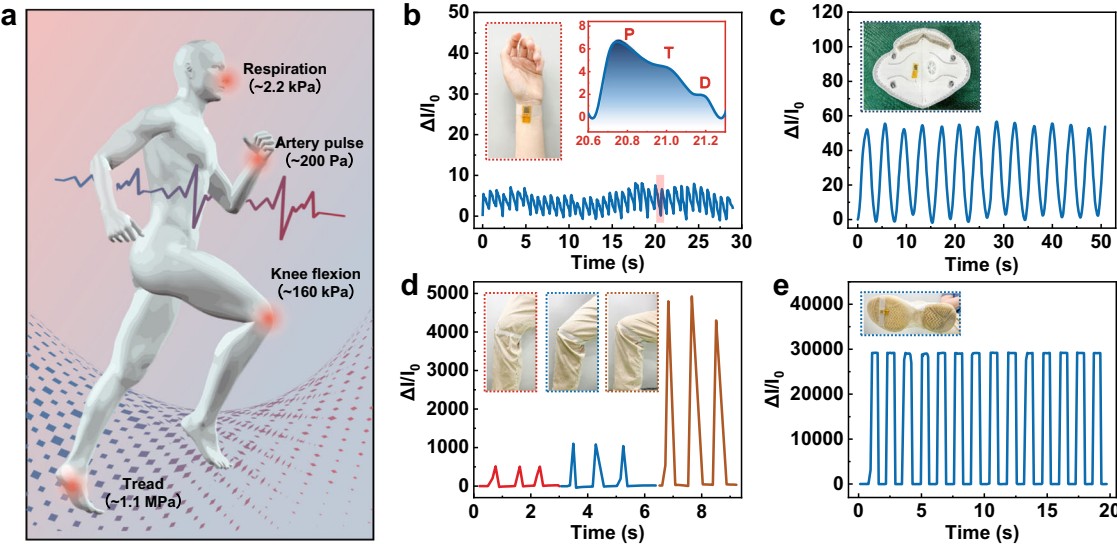

**Fig. 4 | Detection of various physiological signals with DPyCF@SR sensor. a** Important physiological signals for healthcare monitoring. Detection of **b** artery pulse, **c** respiratory rate, **d** knee flexion, and **e** tread.

the output voltage range of the hardware circuit is 0–3.3 V. We defined a keystroke as "light" if the output voltage is less than a prescribed threshold (e.g., 1.5 V) and "hard" if it is greater than the threshold. Considering the fact that the "light" and "hard" press varies from different users, a training algorithm based on the pressing data obtained from a specific user can be used to determine the threshold for distinguishing the "light" and "hard" presses of the user. The hardware and software components of the code-pressure double encryption lock system consist of a data acquisition circuit equipped with a DPyCF@SR sensor array and a host computer program developed by Unity3D platform, respectively. The pressure data is transmitted to a host computer via serial communication, which is parsed and displayed simultaneously to enable human-machine interaction (Fig. 5c). A password can be preset through a user interface shown in Fig. 5d, where the red buttons denote hard keystrokes and the blue buttons denote light keystrokes. For example, by clicking the buttons of red 2, blue 7, red 4, and blue 6 in succession, a password of 2(hard)-7(light)-4(hard)-6(light) is set. When operating, the program would detect the password input through the DPyCF@SR-based keypad. If the signals received are 2(light)-7(hard)-4(hard)-6(light) (Fig. 5e and Supplementary Movie 7), the lock cannot be unlocked although the code numbers are correct. Only when the received signals are 2(hard)-7(light)-4(hard)-6(light) (Fig. 5f and Supplementary Movie 7), which means both code numbers and the stroke pressures match with the preset, the lock is unlocked. Such code-pressure double encryption endows the lock with higher-tier security in comparison with traditional keypad locks.

## Discussion

Traditional flexible pressure sensors can barely achieve high sensitivity and wide linear range together. Here, we tackled this problem by adopting a sensing layer with nonlinear piezo-resistivity and a stiffness regulator with nonlinear elasticity. By synergizing the piezoresistive nonlinearity of the sensing layer and the elastic nonlinearity of the stiffness regulator, high-sensitivity and wide-linear-range pressure sensing are achieved simultaneously. The sensor we developed exhibits an unprecedentedly high sensitivity (24.6 kPa$^{-1}$) over an ultra-wide linear range (1.4 MPa), excellent linearity ($R^2 = 0.999$), and outstanding mechanical durability (50,000 cycles). The great application potential of our sensor in tactile sensing for robotic manipulation, healthcare monitoring, and human-machine interaction has been demonstrated. Our design strategy for achieving both high sensitivity and wide linear

range, which is the synergy of the nonlinearities in mechanical and electrical behaviors, can be further extended to the other types of pressure sensors such as piezocapacitive sensors (Supplementary Note 3).

## Methods

### Fabrication of the sensing layer

On a melamine foam (MF, Sihang Nanotechnology Co., China) block ($20 \times 20 \times 4$ mm$^3$, 32 mg cm$^{-2}$), a $12 \times 12$ micro-pyramids array was produced on both sides by carving with an infrared picosecond laser (PINE-1064-20, China). Then, the pyramidal foam was cleaned by ultrasonication for 10 min in deionized water. After drying, the pyramidal foam was placed into a tubular furnace and heated from room temperature to 650 °C at a rate of 5 °C/min under the atmosphere of 99.999% nitrogen and kept for 30 min for pyrolysis. After cooling down to room temperature in the furnace, a double-sided pyramidal carbon foam array was obtained and sliced into $7 \times 7$ mm$^2$ pieces to be used as the sensing layers of the pressure sensor (Supplementary Fig. 1).

### Fabrication of the stiffness regulator

Parts A and B of the Ecoflex rubber (Smooth-On Ecoflex 0030, USA) were mixed in 1:1 by weight in a petri dish. Then, the mixture was left for 10 min to degas completely before curing at 70 °C for 5 h. After that, the cured rubber with a thickness of approximately 0.8 mm was peeled off from the petri dish and cut into a square frame (outside dimensions $9 \times 9$ mm$^2$, inside dimensions $7 \times 7$ mm$^2$) with laser cutting, which is to be used as a stiffness regulator.

### Fabrication of the DPyCF@SR electrode and packaging of the DPyCF@SR sensor

To fabricate the DPyCF@SR electrode, an $8 \times 8$ cm$^2$ polyethylene terephthalate (PET, Dongguan Hengjie Plastic Raw Material Co., Ltd, China) film was cleaned with deionized water and dried in a drying oven. A 20 nm thick titanium adhesive layer and a 30 nm thick gold layer were sputtered on the PET film sequentially with a magnetron sputtering apparatus (EXPLORER-14, USA). Then, the gold layer was patterned with a 355 nm ultraviolet laser beam (SEAL-355-10S, JPT, China). After that, the PET substrate with the patterned gold layer was cut into small pieces using laser cutting, which will be used as electrodes. Finally, the as-prepared electrode, the double-sided pyramidal carbon foam array (sensing layer), and the stiffness regulator were

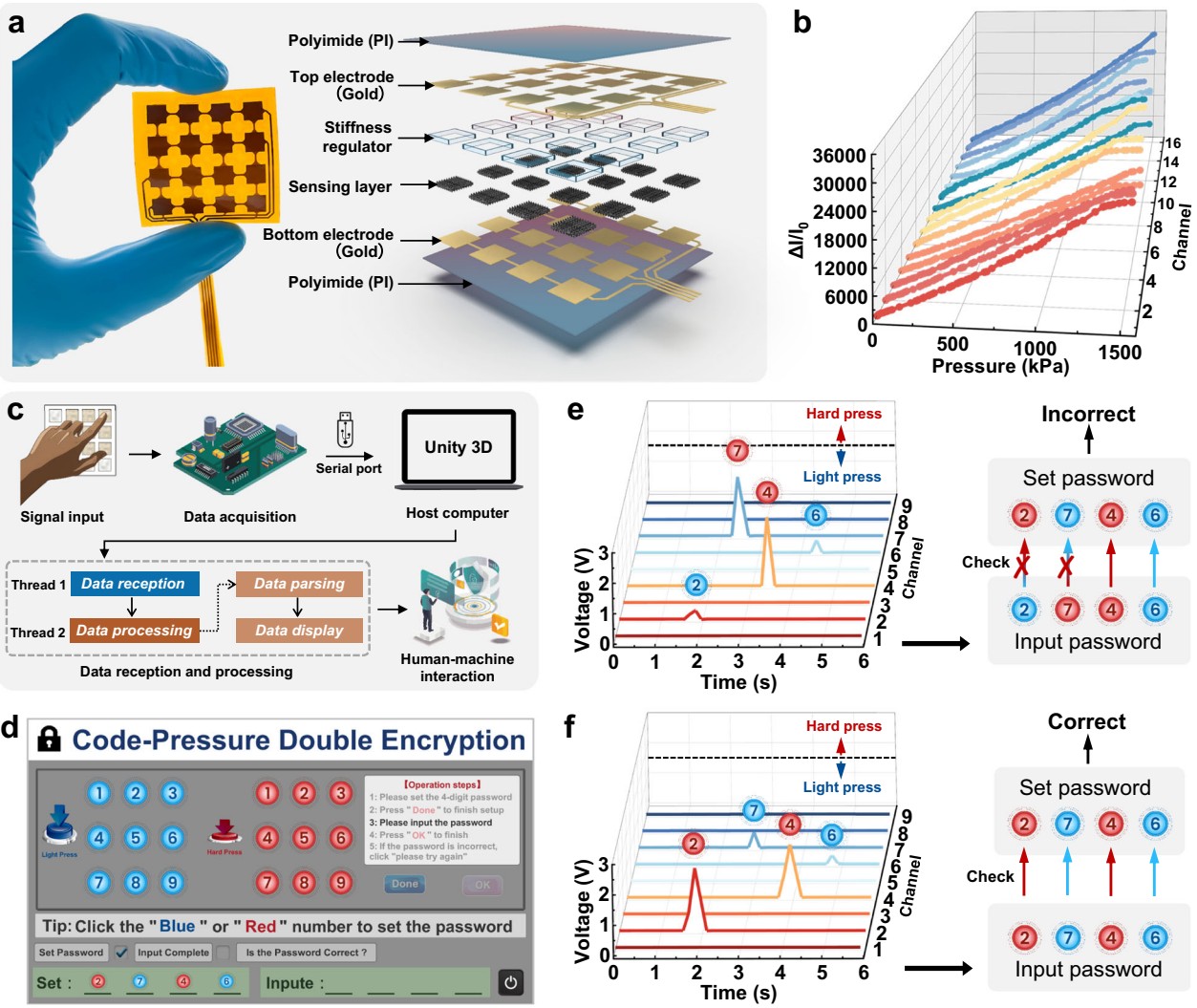

**Fig. 5 | Code-pressure double encryption lock based on DPyCF@SR array. a** The photo and explosion diagram of the DPyCF@SR sensor array. **b** The relative current changes of a DPyCF@SR sensor array under applied pressure. **c** Schematic diagram showing the code-pressure double encryption lock system. **d** User interface of the code-pressure double encryption lock based on Unity3D. **e**, **f** A unsuccessful trial and a successful attempt to unlock the lock.

assembled and packaged with 50 μm thick PI tape (3 M 7413D, USA), resulting in a DPyCF@SR sensor.

## Characterizations and measurements

A scanning electron microscope (SUPRA55 SAPPHIRE, Zeiss Corporation, Germany) equipped with an energy dispersive spectrometer (EDS) was used to characterize the structure and morphology of the MF and DPyCF. A confocal Raman spectrometer (lDSPeC ARCTIC; 532 nm laser wavelength, 50x objective lens) was used to characterize the structural characteristics of the DPyCF. An X-ray diffractometer (XRD, XRD-7000X) with Cu kα radiation (λ = 0.15406 nm) was used to identify the phase structure. The pressure applied to sensors during tests was controlled by a motorized motion platform (FUYU, FLS40, China) with a motion controller (FUYU, FSC-2A, China). Pressure acquisition was done via a parallel beam pressure transducer (HY, HYPX-017, China). The electrical resistance of the DPyCF@SR was measured by a digital source meter (Keithley, 2400, USA).

## Robotic manipulation application

A motorized robotic arm (Universal Robots, UR5, Denmark) was combined with a servo-actuated robotic gripper (ChangingTek, CTM2F110, China) for grasping selected objects including a steel block and a tofu block. The gripper was equipped with a real-time pressure sensor that transmitted data to the host computer via serial communication. The movement of the robotic arm and gripper was determined using a sensor-guided closed-loop control algorithm. To maintain consistency in manipulation, the sensors (DPyCF@SR or Tekscan sensors) were placed beneath a piece of scotch tape (3 M Scotch 810, USA) and then attached them together to the gripper's surface. The frictional coefficient between the scotch tape surface and the objects for the grasping test was measured to be 0.15 for the steel block and 0.04 for the tofu block.

## Fabrication of the sensor array

To develop upper and lower flexible electrodes for the sensor array, a 12.5 μm thick copper layer was patterned on a PI film and partially covered by a 25 μm thick insulation layer. The uncovered areas of the copper layer (4 × 4 array of square patches) can form conductive electrical contacts with the pressure-sensing layer when they are sandwiched. The DPyCF array and the SR for the sensor array were fabricated as previously described. Then, screen printing was used to pattern a glue layer on the insulation layer. Finally, the DPyCF array, the SR, and flexible printed circuits with the glue layer were

laminated together to form the DPyCF@SR sensor array (Supplementary Fig. 22).

## Human research participants

Three males aged 23–29 participated in the physiological signals monitoring and high-resolution pressure sensing study. The purposes and significances of the experiment were informed to the participants before the survey. Participants provided informed consent before the experiment.

## Ethics statement

All procedures during the testing of human participants are approved by the Medical Ethics Committee of Xiamen University. The informed consent of all participants was obtained prior to inclusion in this study.

## Reporting summary

Further information on research design is available in the Nature Portfolio Reporting Summary linked to this article.

## Data availability

The data that support the findings of this study are available from the corresponding authors upon request.

## Code availability

The codes generated in this study are available from the corresponding authors upon request.

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

## Acknowledgements

We acknowledge support from the National Natural Science Foundation of China (no. 52325507, no. U21A20136, no. 52205606) and Major Science and Technology Program of Xiamen City (no. 3502Z20231009).

## Author contributions

H.Y. and W.Z. supervised the research. R.C. designed and performed the experiments. R.C., J.W., and R.W. carried out the device fabrication and the performance measurement. H.Y., R.C., T.L., Y.X., L.Q., and J.W. performed the theoretical analysis. R.C., C.Z., and J.W. analyzed the data. H.Y., R.C., T.L., and W.Z. drafted the manuscript, and all authors contributed to the writing of the manuscript.

## Competing interests

The authors declare no competing interests.
