## [Peer Review File · Nature Communications]

REVIEWER COMMENTS

Reviewer #1 (Remarks to the Author):

In this work, the authors prepared the flexible pressure sensor with a broad detection range and high sensitivity. The linearity is promised by optimizing the resistive property of the sensing layer and the elastic property of the spacer. This device was successfully demonstrated for wearable healthcare monitoring, human-machine interaction, and intelligent controls, etc. I appreciate the design of “nonlinearity synergy” for sensing improvement, however, the wearable demonstrations cannot well explore the significance of this work because the related demonstrations were also achieved by many reports. Apart from the sensing capability, I also recommend the authors to further explore the significant points or unique advantages of this study. My further comments are as below.

- In Supplementary Figure 5, the aspect ratio is important to regulate the decay constant. However, the model that can theoretically support this experimental result is missing. Please provide related analysis that can predict the trend of decay constant based on different aspect ratios.

- In Supplementary Table 1, the fabrication methodology should also be included. There are many reports that can achieve a relatively large detection range and high sensitivity. The competitiveness of this study might be weakened when considering the preparation method and the cost. Please clarify and discuss this concerns into more details.

- As shown in Supplementary Figure 2, the sensing layer is composed of porous pyramidal structures. Is the porosity important for the linearity and sensitivity optimization? Also, how about the structural integrity if the device was exposed to cyclic high pressures? SEM images of the structure after periodical compression at high-pressure range can help to figure out this concern.

- In Supplementary Figure 10b, the loading of 1 Pa can be detected by the sensor with stepwise signal change. Please provide a cyclic test (such as Supplementary Figure 10a) under this low pressure to convince the reliability for small pressure detection. By the way, what is the absolute value of the electrical current when 1 Pa was applied to the device? The absolute values of the initial current and resultant current should be provided alongside to reveal the low-pressure accuracy.

- For the data in Figures 2d-e, 4b-e, S10a-b, etc., the y-axis is pressure instead of relative current variation (as in Figures 2a-c). Consequently, it is difficult to relate the current change with the pressure that has been applied to the device. I would suggest to consolidate the format of related data, so that the readership can relate the signal variation and mechanical stimuli from the plots.

- In the demonstration of robotic manipulation, the successful grasping is also dependent on the surface frictional coefficient between the sensor and the object. What are the frictional coefficients of the DPyCF@SR device and the Tekscan sensor? If the coefficient of the Tekscan sensor is increased, can it also provide similar function as the DPyCF@SR sensor? Furthermore, what is the modulus and weight range of the object that can be successfully manipulated if the robotic arm is assisted by the DPyCF@SR sensor? Please clarify.

- For the wearable sensing in Figure 4d, the device was attached to the knee joint for medium pressure detection. Considering the small size of the sensing layer (7*7 mm²), how to ensure that the device was attached to the joint position that can accurately reflect the bending angle? It can be expected that the sensor will provide different feedback even under the same bending angle, if it was attached to different locations of the knee joint.

- In Figure 4e, the authors indicated that the successful monitoring of sole pressure could be applied as pedometer or tread monitoring. This behavior indeed can also be realized by many other wearable sensors. Because the developed device owns a broad linearity, could it detect the weight change when the human was carrying objects with continuously increased/decreased mass? If so, the real-time electrical signal variation can provide a significant hint to indicate the imposed pressure from the human. Related demonstrations are recommended to reveal the unique capability of this work.

- The author mentioned that the light or hard pressure can help to improve the security when compared with previous code encryption (Figure 5). However, the definition of “light” and “hard” press varies significantly from different users. From this perspective, how to ensure the reliability and practicability of this concept for real applications?

Reviewer #2 (Remarks to the Author):

In this manuscript, the authors proposed a sensitive pressure sensor with ultrawide linear sensing range by balancing both nonlinear piezoresistivity of the sensing layer and nonlinear elasticity of the stiffness regulator. Compared to previously reported literature aiming at balancing sensing performance between high sensitivity and wide linear range (npj Flex. Electron. 2022, 6, 62; Adv. Funct. Mater. 2019, 29, 1902484; Nano-Micro Lett. 2020, 12, 159; Adv. Electron. Mater. 2023, 2201304; ACS Nano 2022, 16, 3, 4338.), this work achieves a satisfactory balance and improvement of these two conflicting performance. Although the authors propose such an ingenious design strategy to simultaneously endow piezoresistive sensors with high sensitivity and wide linearity range, the paper only concludes from theoretical analysis that synergistic electrical and mechanical nonlinear behavior can guide to balance these two contradictory sensing performances. For the key part of the principle, the relevant

characterization is obviously insufficient and the persuasive power seems to be a bit weak. Moreover, in terms of applications, the application scenarios shown in the article do not reflect the uniqueness and irreplaceability of sensors compatible with these two high performances. It gives the impression that other sensors with similar performance as mentioned above seem to be easily functionalized for the above applications. Taking all things into consideration, I think the quality of the article is not up to the level and requirements of Nature Communications in the current stage. Here are my detailed comments.

Comment 1:

Does this nonlinear behavior synergistic strategy reduce the lower detection limit of the sensor device or the detection capability for extremely weak stimuli? From Fig. 2a, we can easily find that although the linear range is wider with the introduction of the stiffness regulator, the sensitivity is correspondingly reduced. Therefore, is this design strategy of coupling mechanical and electrical conditioning enhancing the linear sensing range at the expense of sensitivity?

Comment 2:

As shown in Figures 1c and 2c, the relative current change of this sensor suddenly decreases (i.e., the sensitivity decreases) after the applied pressure reaches the MPa level. Is the reason for this decrease in performance due to damage to the sensing layer's structure? Also, does the sensor maintain a stable electrical output and structural integrity when pressure cycles at the MPa level are applied?

Comment 3:

In my opinion, a description of the approaches that have been reported to balance these two conflicting properties and an analysis of their shortcomings should be added in the Introduction part to highlight the advanced design principles used in this work.

Comment 4:

For the generality of this strategy of synergistic electrical and mechanical nonlinearity to achieve sensors with both high sensitivity and wide linearity range as mentioned in the paper:

- (1) Is it applicable to other types of pressure sensors (e.g., capacitive, piezoelectric principles.)?
- (2) In addition to the separate structures of sensing layer and stiffness regulator designed in the paper, can the above design principles be followed for conductive polymer films with Janus configuration?
- (3) Since the electrical or mechanical nonlinear behavior mentioned in the paper is a function of strain (ϵ), can the above strategy be used to guide the design of strain sensors with both high sensitivity and wide linearity range?

Comment 5:

Here are some minor issues, as mentioned in the article "Parts A and B of the Ecoflex rubber (Smooth-On Ecoflex 0300, USA) were mixed in 1:1 by weight in a petri dish." Is the model number of Ecoflex right here? Maybe Ecoflex 0030 is the correct one.

Reviewer #3 (Remarks to the Author):

The authors presented a novel design of pressure sensor to achieve high sensitivity and large linear range. By investigating the nonlinear resistance change of pyramid carbon foam and nonlinear mechanical response of pyramid carbon foam with Ecoflex regulator, the authors found a synergy between these two properties to achieve linear pressure sensing over a large pressure range. Such pressure sensor was applied to robotic tactile sensing, human physiological monitoring and pressure-code encryption of keyboard. The study is interesting and results are promising. If the authors can fully address the following comments, I'd recommend the publication of the manuscript.

1. The manuscript stated that the sensor array was "nested" by the Ecoflex stiffness regulator, more details should be given on how these two components are fabricated and integrated? Is the SR only a wall at the outer boundary of the sensing pyramid array, or the ecoflex is filling the space among the pyramids?
2. From my reading of SI Fig. 5, I think it's more appropriate to say the decay constant α is between 0.12 and 0.22.
3. In Eq. 3, what's E_0 ? is it of ecoflex or of the combined ecoflex and sensing layer?
4. what's the modulus of ecoflex? what's the mechanical behavior of the pyramidal foam under compression? typically Ecoflex has a Young's modulus of few hundred kPa, when 1.2MPa pressure applied, what's the compressive strain in the pressure sensor and ecoflex SR? How would the deformation affect the structural integrity of the foam? any plastic deformation? Mechanical simulation about the these key components is critical in understanding their properties and performance.
5. In Fig. 4, mark P, T and D waves clearly
6. For wearable applications, mechanical stretching is usually unavoidable, how would the sensor decouple in-plane stretching and out-of-plane pressure, especially when large mechanical stretching is induced?
7. The pyramid dimensions before and after pyrolysis should be given
8. the suppliers of materials should be given, including PET films and PI tape.
9. It's recommended to give schematic illustrations of the fabrication of sensor array, electrode and sensors.

**Response to reviewers' comments on manuscript "Nonlinearity synergy: An elegant strategy for realizing high-sensitivity and wide-linear-range pressure sensing"
(reference number: NCOMMS-23-15442A)**

Reviewer #1:

In this work, the authors prepared the flexible pressure sensor with a broad detection range and high sensitivity. The linearity is promised by optimizing the resistive property of the sensing layer and the elastic property of the spacer. This device was successfully demonstrated for wearable healthcare monitoring, human-machine interaction, and intelligent controls, etc. I appreciate the design of "nonlinearity synergy" for sensing improvement, however, the wearable demonstrations cannot well explore the significance of this work because the related demonstrations were also achieved by many reports. Apart from the sensing capability, I also recommend the authors to further explore the significant points or unique advantages of this study. My further comments are as below.

Response:

We thank the reviewer for the positive comment on our work saying that "*I appreciate the design of "nonlinearity synergy" for sensing improvement...*". By following the reviewer's constructive suggestion of a better demonstration for showing the unique advantages of our DPyCF@SR sensor, we designed and carried out a new experiment, in which our DPyCF@SR sensor was used to detect the variation of the load weight in a sedan. The result showed that our DPyCF@SR sensor can successfully detect a very small change (~1.5 kg) of the load weight in the sedan that weights ~1.5 tons. Such a high-resolution pressure sensing at a highly pressured state cannot be achieved by most of the traditional pressure sensors, reflecting the unique advantages of our sensor in having both high sensitivity and wide linear range. Corresponding revision has been made in the revised manuscript (see the section of "**Performance of pressure sensing**" (Pages 9-10) and Supplementary Video 1 for details).

1. In Supplementary Figure 5, the aspect ratio is important to regulate the decay constant. However, the model that can theoretically support this experimental result is missing. Please provide related analysis that can predict the trend of decay constant based on different aspect ratios.

Our Response:

We thank the reviewer for this constructive comment. By following this suggestion, we developed a theoretical model to predict the dependence of the decay constant (α) on the aspect ratio of the pyramidal structure (λ). The details of the modelling has been added as the Supplementary Note 1 as replicated below

Supplementary Note 1 | Modelling the effect of aspect ratio of DPyCF on the decay constant of electrical resistance

Since the DPyCF sensing layer is a patterned structure composed of repetitive unit cells (see Supplementary Fig. 7a), the overall electrical resistance of the DPyCF sensing layer is given by $R = R_{uc}/N$, where R_{uc} represents the resistance of a unit cell and N is the number of the unit cells. Such a proportionality between R and R_{uc} indicates that they must evolve in the same way with the compression strain, or in other words,

share the same decay constant (α). Therefore, in the following we only focus on the determination of the decay constant of R_{uc} .

For a unit cell of DPyCF (see Supplementary Fig. 7a), the electrical resistance varies with the compressive strain it undergoes. Due to the symmetry about the middle plane, we just need to consider a half of the unit cell, which is composed of a pyramid and a prismatic foundation of area A_2 and thickness h_2 (Supplementary Fig. 7b). To avoid the computational complexity involving the tapering pyramid, we further simplify it by a prism with a smaller cross sectional area of A_1 and height of h_1 (see Supplementary Fig. 7c). The aspect ratio of the DPyCF structure can be roughly correlated with A_1 and h_1 through $\lambda \cong h_1/\sqrt{A_1}$.

Under a compressive force F (see Supplementary Fig. 7d), the unit cell contracts by:

$$\Delta = \delta h_1 + \delta h_2 \cong 2 \left(\frac{F h_1}{E A_1} + \frac{F h_2}{E A_2} \right) \quad (S1)$$

where E is the effective elastic modulus of the DPyCF (porous carbon foam). The average compressive strain of the unit cell is given by:

$$\bar{\varepsilon} = \frac{\Delta}{2(h_1 + h_2)} = \frac{F(A_1 h_2 + A_2 h_1)}{E A_1 A_2 (h_1 + h_2)} \quad (S2)$$

Based on the Eqs. (S1) to (S2), the compressive strains in two prisms can be expressed in terms of the effective strain $\bar{\varepsilon}$ as:

$$\varepsilon_1 = \frac{\Delta h_1}{h_1} = \frac{A_2(h_1+h_2)}{A_1 h_2 + A_2 h_1} \bar{\varepsilon} = \frac{A_2(h_1+h_2)\lambda^2}{A_2\lambda^2 h_1 + h_1^2 h_2} \bar{\varepsilon}, \quad \varepsilon_2 = \frac{\Delta h_2}{h_2} = \frac{A_1(h_1+h_2)}{A_1 h_2 + A_2 h_1} \bar{\varepsilon} = \frac{(h_1+h_2)h_1}{h_1 h_2 + A_2 \lambda^2} \bar{\varepsilon} \quad (S3)$$

Here the relationship $\lambda \cong h_1/\sqrt{A_1}$ is applied to replace A_1 . For porous conductive material, the electrical resistivity exhibits a strong dependence on the compressive strain it is subjected. Such a strain-dependent resistivity is mainly attributed to the increase of the conductive points under compression, and can be expressed by (*J. Mater. Sci.*, 2011, 46, 3186-3190):

$$\rho = \rho_0 \exp(-\varepsilon/\varepsilon_0) \quad (S4)$$

where ρ_0 is the electrical resistivity at zero strain state and ε_0 is a characteristic strain. For our sample (porous carbon foam), it is determined through experimental measurement and curve fitting that $\rho_0 = 4000 \Omega\text{m}$ and $\varepsilon_0 = 0.3$.

$$R_{uc} = 2 \left[\frac{\rho_1 h_1 (1 - \varepsilon_1)}{A_1} + \frac{\rho_2 h_2 (1 - \varepsilon_2)}{A_2} \right] \quad (S5)$$

where $\rho_1 = \rho_0 \exp(-\varepsilon_1/\varepsilon_0)$ and $\rho_2 = \rho_0 \exp(-\varepsilon_2/\varepsilon_0)$ are the resistivity of two compressed prisms.

The electrical resistance of the unit cell can be expressed in terms of the average strain ($\bar{\varepsilon}$) then is given by:

$$R_{uc} = 2\rho_0 \exp\left(-\frac{A_2(\lambda\sqrt{A_1} + h_2)\bar{\varepsilon}}{\varepsilon_0(A_2\lambda\sqrt{A_1} + A_1 h_2)}\right) \left(1 - \frac{A_2(\lambda\sqrt{A_1} + h_2)\bar{\varepsilon}}{A_2\lambda\sqrt{A_1} + A_1 h_2}\right) \frac{\lambda}{\sqrt{A_1}} \quad (S6)$$

$$+2\rho_0 \exp\left(\frac{-\bar{\varepsilon}(\lambda\sqrt{A_1} + h_2)\sqrt{A_1}}{\varepsilon_0(A_2\lambda + h_2\sqrt{A_1})}\right) \left(1 - \frac{(\lambda\sqrt{A_1} + h_2)\sqrt{A_1}}{A_2\lambda + h_2\sqrt{A_1}}\bar{\varepsilon}\right) \frac{h_2}{A_2}$$

Take $h_2 = 0.5$ mm, $A_1 = 0.25$ mm², $A_2 = 1$ mm², the variation of R_{uc} with the average strain $\bar{\varepsilon}$ as given by Eq. (S6) is shown in Supplementary Fig. 8 for different aspect ratio $\lambda = 0.25, 0.5, 1, \text{ and } 2$. In all cases, R_{uc} exhibits a decaying behaviour with increasing $\bar{\varepsilon}$, which can be perfectly fitted by an exponential function $R_{uc}(\bar{\varepsilon}) = R_{uc}^0 \exp(-\bar{\varepsilon}/\alpha)$, where R_{uc}^0 stands for the resistance at zero strain and α is the decay constant. The dependence of the decay constant (α) on the aspect ratio (λ) is shown in Supplementary Fig. 9, which agrees well with the decay constant of the DPyCF sensing layer obtained by experimental measurement and curve fitting (see Supplementary Fig. 6).

Supplementary Fig. 7 | Schematic of the theoretical model for revealing the effect of DPyCF aspect ratio on the decay constant of the electrical resistance. **a** The cross-sectional image of a DPyCF sensing layer showing the patterned structure composed of repetitive unit cells. **b** A half of the unit cell with a pyramid and a prismatic foundation of area A_1 and thickness h_1 . **c** Schematic diagram of the simplified model. **d** Schematic diagram showing the deformed configuration of the model under compression by a force F .

Supplementary Fig. 8 | Variations of the theoretically predicated electrical resistance of a unit cell (R_{uc}) with the subjected compressive strain ($\bar{\epsilon}$) for DPyCF of different aspect ratios (λ). Such decay of the resistance with the compressive strain is fitted with the an exponential function $R_{uc} = R_{uc}^0 \exp(-\bar{\epsilon}/\alpha)$, where the fitting parameter α is called decay constant. a $\lambda = 0.25$, b $\lambda = 0.5$ c $\lambda = 1$, and d $\lambda = 2$.

Supplementary Fig. 9 | Theoretically predicted dependence of the decay constant (α) on the aspect ratio (λ) of the DPyCF structure.

2. In Supplementary Table 1, the fabrication methodology should also be included. There are many reports that can achieve a relatively large detection range and high sensitivity. The competitiveness of this study might be weakened when considering the preparation method and the cost. Please clarify and discuss this concerns into more details.

Our response:

We thank the reviewer for this valuable suggestion. The information of fabrication methodology has been added in the revised Supplementary Table 1.

Compared to the other works, our pressure sensor has competitiveness in not only performance but also cost. Our estimation (see the table R1 below for the details) shows that the fabrication cost of our pressure sensor is about 1 USD/piece, which is much lower than the commercial pressure sensor in the market (e.g., Tekscan FlexiForce A101 Sensor, 10 USD/piece).

Table R1 | Manufacturing costs of DPyCF@SR sensor

	Items	Price of material or processing services	Consumption	Cost per piece
Material	Melamine foam (MF)	10 USD/1 m ³	6.25 cm ³ /pc	0.0000625 USD/pc
	Nitrogen (N ₂)	1 USD/150 L	1.2 L/batch (10 pc)	0.0008 USD/pc
Processing	Magnetron sputtering	20 USD/3000 cm ²	3 cm ² /pc	0.02 USD/pc
	Laser processing	5 USD/h	6 pc/h	0.8 USD/pc
	Pyrolysis*	0.3 USD/h	6 h/batch (10 pc)	0.18 USD/pc
Total cost				1 USD/pc

Note:

* The pyrolysis process lasts for 6 hours each time and 10 samples can be processed each time.

3. As shown in Supplementary Figure 2, the sensing layer is composed of porous pyramidal structures. Is the porosity important for the linearity and sensitivity optimization? Also, how about the structural integrity if the device was exposed to cyclic high pressures? SEM images of the structure after periodical compression at high-pressure range can help to figure out this concern.

Our response:

We thank the reviewer for these valuable comments. Yes, the porosity of the sensing layer (DPyCF) is important for linearity and sensitivity optimization. First, it is well-known that porous structure, such as foam, has much lower effective stiffness and therefore higher deformability compared to the solid counterparts. The pressure sensor based on porous piezoresistive materials should exhibit high sensitivity. That is why porous conductive materials are often used as the sensing layer for high-sensitivity pressure sensing (*Adv. Funct. Mater.* 2020, 39, 2003491; *Small* 2019, 45, 1903487; *Adv. Mater.* 2021, 44, 2104107). For excellent linearity, our study applied the synergy between the nonlinear piezoresistivity of the porous sensing layer (DPyCF) and the nonlinear elasticity of the stiffness regulator (SR). The piezoresistive nonlinearity of the sensing layer depends on the porosity of the carbon foam and other structural features,

such as the aspect ratio of the pyramidal structures.

Our DPyCF has high deformability and can withstand significant compression and recover instantly. To further examine its integrity under cyclic high pressure, we performed fatigue test with cyclic pressure of 1 MPa in magnitude. The results (Supplementary Fig. 14) show that the DPyCF sensor can output an undamped electrical signal under cyclic pressure load, implying no loss of structural integrity. Moreover, we performed SEM imaging on the DPyCF sensing layer after the cyclic test (~7800 cycles). The results show that the pyramidal porous structure exhibited no big change (Supplementary Figs. 15a) as compared to the structure before test (Supplementary Figs. 15b). No apparent collapse or breakage of the porous structure or skeleton framework is observed.

Supplementary Fig. 14 | Output (relative change of current) of a DPyCF@SR sensor under a cyclic pressure load with amplitude of 1 MPa and approximately 7,800 cycles.

Supplementary Fig. 15 | Scanning electron microscopy (SEM) of the DPyCF before and after high-pressure cyclic compression. **a** SEM images of the DPyCF after high-pressure (1 MPa) cyclic compression. **b** SEM images of the DPyCF before the compression test.

4. In Supplementary Figure 10b, the loading of 1 Pa can be detected by the sensor with stepwise signal

change. Please provide a cyclic test (such as Supplementary Figure 10a) under this low pressure to convince the reliability for small pressure detection. By the way, what is the absolute value of the electrical current when 1 Pa was applied to the device? The absolute values of the initial current and resultant current should be provided alongside to reveal the low-pressure accuracy.

Response:

We thank the reviewer for this valuable comment. To carry out the low-pressure cyclic test, we developed an experimental setup as schematically shown in Supplementary Fig. 17a, through which a cyclic load around 1 Pa can be applied to the sensor under test. The test results are shown in Supplementary Fig. 17b, displaying the excellent capability of our sensors in detecting small pressure under cyclic loading condition. The absolute value of the initial current (I_0) was 2.228 μA at 10V test voltage. The resultant current change was ~ 60 nA when 1 Pa of pressure was applied.

Supplementary Fig. 17 | Ultra-low cyclic pressure test. **a** Schematic diagram showing the setup for ultra-low cyclic pressure test. The force causing the bending of the polyimide (PI) film is equally applied to the DPyCF@SR sensor and the commercial high-precision force sensor (SMT-200N-S, AiLogics, USA). **b** The output (current change) of the DPyCF@SR sensor agrees well with the output of the commercial force sensor, reflecting the high performance of the DPyCF@SR sensor in detecting cyclic lower pressure.

5. For the data in Figures 2d-e, 4b-e, S10a-b, etc., the y-axis is pressure instead of relative current variation (as in Figures 2a-c). Consequently, it is difficult to relate the current change with the pressure that has been applied to the device. I would suggest to consolidate the format of related data, so that the readership can relate the signal variation and mechanical stimuli from the plots.

Response:

We thank the reviewer's constructive comment. We have updated changed the y-axis in Figs. 2d-e, 4b-e, and S10a-b (revised Supplementary Figs. 16 and 17) to the relative current variation accordingly.

6. In the demonstration of robotic manipulation, the successful grasping is also dependent on the surface frictional coefficient between the sensor and the object. What are the frictional coefficients of the

DPyCF@SR device and the Tekscan sensor?

Response:

We thank the reviewer for this valuable comment. In our grasping tests, to avoid the influence of the sensor's surface on the friction, we placed the tested sensor (DPyCF@SR or Tekscan sensors) beneath an adhesive tape (3M Scotch 810, USA) and then attached them together to the gripper's surface. By doing so, the tape surface, rather than the sensor's surface, was in direct contact with the tested sample (e.g., steel block or tofu block). Our experimental measurements showed that the frictional coefficient between the tape surface and the steel block is around 0.15, while ~0.04 for the tofu block. To clarify this point, some discussions have been added in the "**Robotic manipulation application**" section of **Methods** (see Line 345, Page 19).

If the coefficient of the Tekscan sensor is increased, can it also provide similar function as the DPyCF@SR sensor?

Response:

If the coefficient of friction between the tape covering the Tekscan sensor and the steel block is increased from 0.15 to 0.3, our calculation showed that the grasping pressure to lift a 900 g steel block is less than 375 kPa, which falls in the detection range of the Tekscan sensor. Therefore, the success of the lifting test is expected. But, the benefit of increasing the coefficient of friction is limited since it cannot be increased infinitely and meanwhile the surface roughness induced for higher friction may also affect the sensing accuracy of the normal pressure.

Furthermore, what is the modulus and weight range of the object that can be successfully manipulated if the robotic arm is assisted by the DPyCF@SR sensor? Please clarify.

Response:

The success of the grasping and lifting test requires the satisfaction of two conditions: (1) The maximum static friction force between the grippers and the sample should be sufficient to surmount the sample's weight; (2) the squeezing pressure applied to the sample by the grippers should not cause fracture of the sample.

For a sample like the steel block, the condition (2) is automatically satisfied due to the superior mechanical property of steel. Therefore, in theory, the maximum weight of the steel block that can be lifted is determined by the measurement range of the sensor and the maximum squeezing force generated by the robotic arm, whichever is less.

For a fragile sample like the tofu block, satisfaction of condition (2) implies a critical pressure denoted by p_{cr} , beyond which the fracture of the tofu would happen. The value of p_{cr} depends on the fracture strength (rather than modulus) of the fragile sample. The maximum weight of the tofu block that can be lifted is equal to $\mu p_{cr} A$, where μ is the maximum static frictional coefficient and A is the contact area.

7. For the wearable sensing in Figure 4d, the device was attached to the knee joint for medium pressure

*detection. Considering the small size of the sensing layer (7*7 mm²), how to ensure that the device was attached to the joint position that can accurately reflect the bending angle? It can be expected that the sensor will provide different feedback even under the same bending angle, if it was attached to different locations of the knee joint.*

Response:

We thank the reviewer for this valuable comment. We agree with the reviewer that a sensor if attached to different position of the knee will provide different feedback even under the same bending angle. But measuring the accurate absolute bending angle is not our purpose here. Actually, our purpose here is to monitor the bending angle of the knee relative to a reference state based on the change of the current in the sensor attached. To ensure a high accuracy of the measurement, pre-calibration of the sensor is needed after installation to eliminate the discrepancy in the output signal caused by the installation position and approach of the sensor.

8. In Figure 4e, the authors indicated that the successful monitoring of sole pressure could be applied as pedometer or tread monitoring. This behavior indeed can also be realized by many other wearable sensors. Because the developed device owns a broad linearity, could it detect the weight change when the human was carrying objects with continuously increased/decreased mass? If so, the real-time electrical signal variation can provide a significant hint to indicate the imposed pressure from the human. Related demonstrations are recommended to reveal the unique capability of this work.

Response:

Thank the reviewer for this constructive comment. To demonstrate the unique advantages of our DPyCF@SR sensor, we designed and carried out a new experiment, in which our DPyCF@SR sensor was used to detect the variation of the load weight in a sedan (see Supplementary Fig.18). The result (see Fig.2) showed that our DPyCF@SR sensor can successfully detect a very small change (~1.5 kg) of the load weight in the sedan that weights ~1.5 tons. Such a high-resolution pressure sensing at a highly pressured state cannot be achieved by most of the traditional pressure sensors, reflecting the unique advantages of our sensor in having both high sensitivity and wide linear range. Corresponding revision has been made in the revised manuscript (see the section of “**Performance of pressure sensing**” (Pages 9-10) and Supplementary Video 1 for details).

Supplementary Fig. 18 | Experimental setup for detecting the variation of load weight in a sedan weighing ~1.5 tons.

Fig. 2 Sensing performance of the DPyCF@SR pressure sensor. f-h Variation of the relative current that reflects the changes of the load weight in a 1.5-ton sedan (Supplementary Video 1).

9. The author mentioned that the light or hard pressure can help to improve the security when compared with previous code encryption (Figure 5). However, the definition of “light” and “hard” press varies significantly from different users. From this perspective, how to ensure the reliability and practicability of this concept for real applications?

Response:

We thank the reviewer for this valuable comment. We agree with the reviewer that there is a significant diversity among people about the definition of “light” and “hard” press. To ensure the reliability and practicability of this concept in real applications, one can apply a machine learning-based algorithm to train the programme with the data of “light” or “hard” presses input by a user. So that, customized definitions of “light” and “hard” press for a specific user can be determined.

To clarify this point, we have added some discussions in the section of “Code-pressure double encryption” (see Line 264, Page 15).

Reviewer #2

In this manuscript, the authors proposed a sensitive pressure sensor with ultrawide linear sensing range by balancing both nonlinear piezoresistivity of the sensing layer and nonlinear elasticity of the stiffness regulator. Compared to previously reported literature aiming at balancing sensing performance between high sensitivity and wide linear range (npj Flex. Electron. 2022, 6, 62; Adv. Funct. Mater. 2019, 29, 1902484; Nano-Micro Lett. 2020, 12, 159; Adv. Electron. Mater. 2023, 2201304; ACS Nano 2022, 16, 3, 4338.), this work achieves a satisfactory balance and improvement of these two conflicting performance. Although the authors propose such an ingenious design strategy to simultaneously endow piezoresistive sensors with high sensitivity and wide linearity range, the paper only concludes from theoretical analysis that synergistic electrical and mechanical nonlinear behavior can guide to balance these two contradictory sensing performances. For the key part of the principle, the relevant characterization is obviously insufficient and the persuasive power seems to be a bit weak. Moreover, in terms of applications, the application scenarios shown in the article do not reflect the uniqueness and irreplaceability of sensors compatible with these two high performances. It gives the impression that other sensors with similar performance as mentioned above seem to be easily functionalized for the above applications. Taking all things into consideration, I think the quality of the article is not up to the level and requirements of Nature Communications in the current stage. Here are my detailed comments.

Response:

We sincerely thank the reviewer for the positive comment on our work saying that “*propose such an ingenious design strategy to simultaneously endow piezoresistive sensors with high sensitivity and wide linearity range*”. However, we cannot accept the reviewer’s criticism saying that “*the relevant characterization is obviously insufficient and the persuasive power seems to be a bit weak*” because we actually carried out a bunch of characterizations as exemplified below:

(1) We characterized the nonlinear stress-strain relationship (stiffening behavior) of the stiffness regulator (SR) (see Supplementary Fig. 10) and the dependence of the stiffening constant (β) on the mixing ratio of the compositions of the building material (see Supplementary Fig. 11).

(2) We measured the decaying behavior of the electrical resistance of the sensing layer (DPyCF) with the subjected compressive strain (see Supplementary Fig. 5) and the dependence of the decay constant (α) on the aspect ratio of the pyramid structure (see Supplementary Fig. 6). In the revision, we further developed a theoretical model to obtain the insights into such dependence (see Supplementary Note 1).

(3) We characterized the pressure sensing performance of 6 DPyCF@SR sensors with different combinations of decay constant (α) and stiffening constant (β), showing the importance of the synergy between the decay constant (α) and stiffening constant (β) in ensuring the high sensitivity and wide range of the sensors simultaneously (see Supplementary Fig. 13).

To demonstrate the unique advantages and irreplaceability of our DPyCF@SR sensor, we designed and carried out a new experiment, in which our DPyCF@SR sensor was used to detect the variation of the load weight in a sedan. The result showed that our DPyCF@SR sensor can successfully detect a very small change (~1.5 kg) of the load weight in the sedan that weights ~1.5 tons. Such a high-resolution pressure sensing at a highly pressured state cannot be achieved by most of the traditional pressure sensors, reflecting the unique advantages of our sensor in having both high sensitivity and wide linear range. Corresponding revision has been made in the revised manuscript (see the section of “**Performance of pressure sensing**” (Pages 9-10)

and Supplementary Video 1 for details).

1. Does this nonlinear behavior synergistic strategy reduce the lower detection limit of the sensor device or the detection capability for extremely weak stimuli? From Fig. 2a, we can easily find that although the linear range is wider with the introduction of the stiffness regulator, the sensitivity is correspondingly reduced. Therefore, is this design strategy of coupling mechanical and electrical conditioning enhancing the linear sensing range at the expense of sensitivity?

Response:

We appreciate the reviewer's questions. In theory, the introduction of the nonlinear stiffness regulator (SR) will affect the sensitivity of the sensor. In practice, however, such influence is limited since the modulus of the SR is quite small at low pressure load. Thanks to the stiffening behavior of the Ecoflex rubber, the modulus of the SR increases at high pressure load, which greatly broadens the range of measurement, and synergizes with the nonlinear sensing layer (DPyCF) to achieve the high sensitivity and wide-range linearity together. Our DPyCF@SR sensor can detect small pressure (1 Pa) even under cyclic loading condition (see Supplementary Fig. 17), which is beyond the scope of most piezoresistive pressure sensors (see Supplementary Table 1).

2. As shown in Figures 1c and 2c, the relative current change of this sensor suddenly decreases (i.e., the sensitivity decreases) after the applied pressure reaches the MPa level. Is the reason for this decrease in performance due to damage to the sensing layer's structure? Also, does the sensor maintain a stable electrical output and structural integrity when pressure cycles at the MPa level are applied?

Response:

We appreciate the reviewer's questions. The decrease of the relative current is not due to the damage in the sensing layer. Instead, it results from the saturation of the electrical resistance of the sensing layer at a compressive strain above 0.75, as shown in Fig. R1 below.

The DPyCF sensing layer of our sensors exhibits high deformability and can withstand significant compression and recover instantly. To further examine its integrity under cyclic high pressure, we performed fatigue test with cyclic pressure of 1 MPa in magnitude. The results (see Supplementary Fig. 14) show that the DPyCF sensor can output an undamped electrical signal under cyclic pressure load, implying no loss of structural integrity. Moreover, we performed SEM imaging on the DPyCF sensing layer after the test (~7800 cycles). The results show that the pyramidal porous structure exhibited no significant change (see Supplementary Fig. 15a) as compared to the structure before test (see Supplementary Fig. 15b). No apparent collapse or breakage of the porous structure or skeleton framework is observed.

Fig. R1 | The variation of electrical resistance of a double-sided pyramidal carbon foam (DPyCF) with the compressive strain from 0 to 0.8.

Supplementary Fig. 14 | Output (relative change of current) of a DPyCF@SR sensor under a cyclic pressure load with amplitude of 1 MPa and approximately 7,800 cycles.

Supplementary Fig. 15 | Scanning electron microscopy (SEM) of the DPyCF before and after high-pressure cyclic compression. **a** SEM images of the DPyCF after high-pressure (1 MPa) cyclic compression. **b** SEM images of the DPyCF before the compression test.

3. In my opinion, a description of the approaches that have been reported to balance these two conflicting properties and an analysis of their shortcomings should be added in the Introduction part to highlight the advanced design principles used in this work.

Response:

We thank the reviewer for this valuable comment. We have updated the “**Introduction**” section (Pages 2-3) by adding more discussions, as replicated below, on the previous endeavours to address the conflict between high sensitivity and wider measurement range as well as their shortcomings:

“.....To date, various methods have been developed to enhance the performance of the flexible pressure sensor. To enhance the sensitivity of flexible pressure sensors, a variety of surface topological microstructures, such as pyramidal³²⁻³⁴, interlocked^{35,36}, cylindrical³⁷, and domed microstructures³⁸, were employed in the pressure sensing layer. However, sensors based on these surficial microstructures normally exhibit sensitivity lower than 10 kPa⁻¹, which is the minimum sensitivity requirement for tactile sensing in dexterous robotic manipulation⁸. For higher sensitivity, sensing layers with interior microscopic porous structures and therefore notable deformability were adopted in pressure sensors³⁹⁻⁴¹. Nevertheless, the aforementioned strategies, rooted in either surficial topological microstructures or interior microscopic porosity, suffer from a narrow linear range no more than 100 kPa. To extend the linear range, hybrid surficial topographical microstructures, which combine micro-dome and micro-cone arrays, were applied in the sensing layers, resulting in a remarkable extension of the linear range to 1 MPa. However, sensors based on the hybrid surficial topographical microstructures show a sensitivity only around 0.3 kPa⁻¹²³. To reconcile the intrinsic conflict between the high sensitivity and wide linear range in the traditional flexible pressure sensors, sensing layers with a hierarchical microstructure based on a porous lattice structure were applied and proven to be an effective strategy for achieving a moderate sensitivity of 4.7 kPa⁻¹ across a broad linear range of 1 MPa³¹. For a further augment of sensitivity within this ultra-wide linear range of 1 MPa, a hybrid hierarchical structure integrating microscopic gradient pores and pyramidal surficial microstructure was employed within a flexible sensor⁴², yielding a sensitivity surpassing 10 kPa⁻¹. However, fabricating the gradient pores with high controllability in its geometry was proven challenging, which thereby compromises the reproducibility and reliability of this strategy. In addition, the principle governing the efficacy of this strategy remain obscure, making it quite difficult to further improve and optimize this approach.....”

4. For the generality of this strategy of synergistic electrical and mechanical nonlinearity to achieve sensors with both high sensitivity and wide linearity range as mentioned in the paper:

(1) Is it applicable to other types of pressure sensors (e.g., capacitive, piezoelectric principles.)?

Response:

We thank the reviewer for this valuable comment. Our proposed nonlinear synergy strategy is also applicable to other types of pressure sensors such as the capacitive and piezoelectric pressure sensors. To demonstrate this point, we designed and developed capacitive pressure sensors under the guideline of this strategy. The details has been added as the Supplementary Note 3 as replicated below.

Supplementary Note 3 | Applicability of strategy of nonlinearity synergy in capacitive sensors

To verify the applicability of our strategy of nonlinearity synergy in the other types of pressure sensors, we designed and fabricated capacitive sensors by using carbon nanotubes (CNTs)-doped polydimethylsiloxane (PDMS) as the material for pressure sensing layers. The sensing layer is prepared by assembling two pieces of PDMS/CNT sheets with micro-pyramid array on one side in a face-to-face configuration (Supplementary Fig. 21). Then, the as-prepared sensing layer is sandwiched between a pair of electrodes, forming a capacitive pressure sensor. To modulate the capacitive and mechanical properties of the sensing layers, we adopted two different mass fractions of CNTs, 5% and 3%, when preparing the sensing layers. The resultant sensors are named as CNT@5 and CNT@3, respectively.

The capacitances of these two kinds of sensors were tested under a varying compressive strain (ε) (Supplementary Figs. 22a and 22d), showing ever-increasing capacitances (C) of the sensors with the applied compressive strain (ε). The strain dependence of the capacitances can be perfectly ($R^2 > 0.99$) fitted by exponential functions in the form of

$$C = C_0 \exp(\varepsilon/\gamma) \quad (S7)$$

where C_0 is the initial capacitance at zero strain and γ is a constant characterizing the increasing rate of the capacitance with the compressive strain. For CNT@5 and CNT@3, we found $\gamma_1 = 0.459$ and $\gamma_2 = 2.357$ (Supplementary Figs. 22a and 22d), respectively. From Eq. (S7), the relative variation of the capacitance ($\Delta C/C_0$) thus can be expressed as a function of ε :

$$\frac{\Delta C}{C_0} = \frac{C - C_0}{C_0} = \exp(\varepsilon/\gamma) - 1 \quad (S8)$$

On the other hand, the mechanical behaviour of CNT@5 and CNT@3 under compression was also characterized (Supplementary Figs. 22b and 22e). The nominal pressure (p), which is defined as the applied force divided by the area enclosed by the outer perimeter of the sensor, exhibits a clear nonlinear dependence on the compressive strain (ε). Such nonlinear pressure-strain relationships can be perfectly ($R^2 > 0.99$) fitted by an exponential function as:

$$p = \beta E_0 [\exp(\varepsilon/\beta) - 1] \quad (S9)$$

where E_0 is the initial tangential modulus of the sensor at zero strain and β is the stiffening constant. For CNT@5 and CNT@3, we found $\beta_1 = 0.314$ and $\beta_2 = 0.235$, respectively (Supplementary Figs. 22b and 22e). In the light of Eqs. (S8) and (S9), the sensitivity (S) of the sensor can be expressed as:

$$S \equiv \frac{d(\Delta C/C_0)}{dp} = \frac{1}{\beta E_0} \exp\left[\left(\frac{1}{\gamma} - \frac{1}{\beta}\right)\varepsilon\right] \quad (S10)$$

Eq. (S10) implies that the linearity of the capacitive sensor depends on the difference between γ and β , which is similar to what is implied in Eq. (4) for the DPyCF@SR resistive sensors. Ideally, perfect linearity ($R^2 = 1.0$) can be achieved when $\beta = \gamma$. In practice, however, γ and β would not be exactly the same. Under this circumstance, the closer the values of γ and β , the higher the linearity of the sensors. This theoretical prediction from Eq. (S10) was verified by comparing the linearity of CNT@5 and CNT@3. For CNT@5 with $\beta_1 = 0.314$ and $\gamma_1 = 0.459$, a high linearity with $R^2 > 0.99$ was measured in the range of 0-100 kPa (Supplementary Fig. 22c). In contrast, for CNT@3 with $\beta_2 = 0.235$ and $\gamma_2 = 2.357$, a relatively lower linearity ($R^2 < 0.95$) was detected (Supplementary Fig. 22f). It is demonstrated that our strategy of nonlinearity synergy can also be applied to the capacitive pressure sensors for a wider linear range.

Supplementary Fig. 21 | Capacitive pressure sensor made of polydimethylsiloxane (PDMS) doped with carbon nanotubes (CNTs). **a** The schematic diagram showing the design of the capacitive sensor. **b** The photograph of an as-prepared capacitive pressure sensor.

Supplementary Fig. 22 | Characterizations of two capacitive pressure sensors. **a** The variation of capacitance of CNT@5 with varying compressive strain. **b** The compressive pressure-strain curve of the CNT@5. **c** Relative capacitance changes as a function of pressure for the CNT@5. **d** The variation of capacitance of CNT@3 with varying compressive strain. **e** The compressive pressure-strain curve of the CNT@3. **f** Relative capacitance changes as a function of pressure for the CNT@3.

(2) In addition to the separate structures of sensing layer and stiffness regulator designed in the paper, can the above design principles be followed for conductive polymer films with Janus configuration?

Response:

We thank the reviewer for this valuable comment. We think conductive polymer films can function as both stiffness regulators and sensing layers (e.g., porous conductive polymer films). If the nonlinearity in its mechanical and electrical properties are well design, we may also achieve high sensitivity and wide linear range together. This is an interesting idea worthy of further study.

(3) Since the electrical or mechanical nonlinear behavior mentioned in the paper is a function of strain (ϵ), can the above strategy be used to guide the design of strain sensors with both high sensitivity and wide linearity range?

Response:

We thank the reviewer for this inspiring question. Exactly, the electrical resistance of the DPyCF sensing layer (R) and the pressure load (p) of SR are both nonlinear functions of the subjected strain (ϵ), we therefore applied the strain (ϵ) as the intermediate quantity to correlate the pressure (p) and the electrical signal. By coordinating the electrical and mechanical nonlinearities, we successfully achieved the high-sensitivity pressure sensing and wide linear range together. If strain, rather than pressure, is the quantity to be measured, one can design the strain sensor directly by using the strain(ϵ)-resistance (R) relationship of the sensing layer. There is no need to consider the strain(ϵ)-pressure(p) relationship any more. In that case, however, achieving a wide linear range becomes challenging because of the nonlinear relationship between the strain and electrical resistance of the DPyCF sensing layer.

5. Here are some minor issues, as mentioned in the article “Parts A and B of the Ecoflex rubber (Smooth-On Ecoflex 0300, USA) were mixed in 1:1 by weight in a petri dish.” Is the model number of Ecoflex right here? Maybe Ecoflex 0030 is the correct one.

Response:

We thank the reviewer for spotting this typo, which has been corrected in the revised manuscript.

Reviewer #3

The authors presented a novel design of pressure sensor to achieve high sensitivity and large linear range. By investigating the nonlinear resistance change of pyramid carbon foam and nonlinear mechanical response of pyramid carbon foam with Ecoflex regulator, the authors found a synergy between these two properties to achieve linear pressure sensing over a large pressure range. Such pressure sensor was applied to robotic tactile sensing, human physiological monitoring and pressure-code encryption of keyboard. The study is interesting and results are promising. If the authors can fully address the following comments, I'd recommend the publication of the manuscript.

Response:

We thank the reviewer for the positive comment on our work saying that “*The study is interesting and results are promising.*” and for providing valuable insights to further improve this manuscript. Below, we present our point-to-point responses to the specific comments.

1. The manuscript stated that the sensor array was “nested” by the Ecoflex stiffness regulator, more details should be given on how these two components are fabricated and integrated? Is the SR only a wall at the outer boundary of the sensing pyramid array, or the ecoflex is filling the space among the pyramids?

Response:

We thank the reviewer for this valuable comment. To provide a clearer visualization of the nesting details between DPyCF and SR, we have added a schematics (Supplementary Fig. 4), which illustrates the “wall”-like stiffness regulator (SR) around the pyramidal array of DPyCF.

Supplementary Fig. 4 | The schematics of SR and DPyCF nested.

The detailed fabrication process of the DPyCF and SR sensing layer was provided in the “**Fabrication of the sensing layer**” and “**Fabrication of the stiffness regulator**” section of **Methods** (Pages 17-18). A schematic diagram showing the assembling process of the SR, DPyCF and electrode into sensor has been added in Supplementary Fig. 1.

2. From my reading of SI Fig. 5, I think it's more appropriate to say the decay constant α is between 0.12 and 0.22.

Response:

We appreciate the reviewer’s suggestion. We have modified the range of the decay constant in the revised manuscript.

3. In Eq. 3, what's E_0 ? is it of ecoflex or of the combined ecoflex and sensing layer?

Response:

Thanks for the reviewer’s question. E_0 is the tangential modulus of the ensemble of SR and the sensing layer (DPyCF) at zero strain. The corresponding sentences have been revised in the section of the “**Accomplishment of high sensitivity together with a wide linear range**” (see Line 106, Page 5).

4. What's the modulus of ecoflex? what's the mechanical behavior of the pyramidal foam under compression? typically Ecoflex has a Young's modulus of few hundred kPa, when 1.2MPa pressure applied, what's the compressive strain in the pressure sensor and ecoflex SR?

Response:

To evaluate the mechanical behavior of the SR (Ecoflex), we performed a compression test. The stress-strain curve is shown in Figure R2, which shows the nonlinear elasticity of the SR. From the curve, the initial modulus of Ecoflex is estimated to be 59.26 kPa. When 1.2MPa pressure applied, the compressive strain in the pressure sensor and ecoflex SR are close and equal to 0.7.

Fig. R2 | The pressure-strain curve of the SR under compression.

To reveal the mechanical behaviour of the DPyCF under compression, we observed a DPyCF under cyclic compression from the side with an optical microscope (A0-HK830-5870T, China). Figure R3 shows the snapshots of a loading and unloading process. It can be seen that that the DPyCF exhibits remarkable recovery capability. It can quickly return to its initial configuration without any apparent structural damage or plastic deformation, even after being compressed by 70%. Thus, the DPyCF is nonlinear elastic within the compression strain of 70%. Figure R4 shows the stress-strain curve of the pyramidal foam under compression.

Fig. R3 | The optical images of DPyCF at various compressive strains ranging from 0 to 70%.

Fig. R4 | The mechanical behavior of the pyramidal carbon foam (DPyCF).

How would the deformation affect the structural integrity of the foam? any plastic deformation? Mechanical simulation about the these key components is critical in understanding their properties and performance.

Response:

To further examine its integrity under cyclic high pressure, we performed fatigue test with cyclic pressure of 1 MPa in magnitude. The results (Supplementary Fig. 14) show that the DPyCF sensor can output a undamped electrical signal under cyclic pressure load, implying no loss of structural integrity. Moreover, we performed SEM imaging on the DPyCF sensing layer after the test (~ 7800 cycles). The results show that the pyramidal porous structure exhibited no big change (Supplementary Fig. 15a) as compared to the structure before test (Supplementary Fig. 15b). No plastic deformation, collapse or breakage of the porous structure or skeleton framework is observed.

As to the numerical simulation suggested by the reviewer, we think it is not necessary for our study because no damage or plastic deformation of DPyCF has been observed in experiment. The information that can be

obtained from simulation such as stress field seems not so important for our study.

Supplementary Fig. 14 | Output (relative change of current) of a DPyCF@SR sensor under a cyclic pressure load with amplitude of 1 MPa and approximately 7,800 cycles.

Supplementary Fig. 15 | Scanning electron microscopy (SEM) of the DPyCF before and after high-pressure cyclic compression. **a** SEM images of the DPyCF after high-pressure (1 MPa) cyclic compression. **b** SEM images of the DPyCF before the compression test.

5. In Fig. 4, mark P, T and D waves clearly.

Response:

Thanks for the reviewer's suggestion. P, T, D waves have been marked in the revised Fig.4b.

6. For wearable applications, mechanical stretching is usually unavoidable, how would the sensor decouple in-plane stretching and out-of-plane pressure, especially when large mechanical stretching is induced?

Response:

We thank the reviewer for the valuable comment. We performed stretching tests on the DPyCF@SR sensor to evaluate the effect of in-plane stretching on its electrical signal output. The results show that the rate of current change less than 0.05 when the tensile strength increase to 200 Nm^{-1} (Fig. R4). The effect of in-plane stretching on the electrical signal of the sensor is very small and negligible compared to out-of-plane stretching. Therefore, in-plane and out-of-plane pressure do not interfere with each other.

Fig. R4 | The relative current changes as a function of pressure on DPyCF@SR sensors with different tensile strength.

7. The pyramid dimensions before and after pyrolysis should be given

Response:

We thank the reviewer for this suggestion. The pyramid dimensions before and after pyrolysis have been added in the updated Supplementary Fig. 3.

8. the suppliers of materials should be given, including PET films and PI tape.

Response:

We thank the Reviewer for this suggestion. We have added the suppliers information to the revised manuscript (Pages 17-18).

9. It's recommended to give schematic illustrations of the fabrication of sensor array, electrode and sensors.

Response:

We sincerely thank the reviewer for this suggestion. We have updated the fabricating process flow diagram in Supplementary Fig. 1 by incorporating the SR fabrication process and the electrode fabrication process. We have added Supplementary Fig. 20 to illustrate the fabrication process of the sensor array.

Supplementary Fig. 1 | Fabrication processes of the DPyCF sensor. a Fabrication processes of the sensing layer. Firstly, a 3D model of the pyramidal array is created with CAD software (SoildWorks), and then the 3D model is sliced for trajectory planning of the laser processing. Subsequently, a planned processing trajectory is introduced into an infrared picosecond laser, and then the melamine foam (MF) is manufactured using the 3D dynamic focusing technique of the laser to obtain an MF-based double-sided pyramidal array. Finally, the MF-based double-sided pyramidal array is pyrolyzed at a high temperature to produce conductive double-sided pyramidal carbon foam (DPyCF). **b** Fabrication processes of the stiffness regulator. **c** Fabrication processes of the electrode.

Supplementary Fig. 20 | Fabrication processes of the DPyCF@SR sensor array. First, flexible circuits are designed using electronic design automation (EDA) software. Next, the designed circuits are fabricated into flexible circuit boards (FPCBs) through machining processes. Subsequently, screen printing is conducted on the surface of the FPCB, and glue is applied to the electrode side of the FPCB. The machined DPyCF and SR components are then assembled onto the FPCB electrodes. Finally, another FPCB is tightly assembled face-to-face with the FPCB containing the pressure-sensitive layer, completing the fabrication of the DPyCF sensor array.

REVIEWERS' COMMENTS

Reviewer #1 (Remarks to the Author):

The authors have provided experiments and sufficient revisions to address my previous concerns. The working mechanism is now more clear when compared with the last version. Also, the sensing demonstration can well prove the significant advance of the sensor design for a wide working range and linearity.

I have no more scientific and technological comments at this stage and I would like to recommend the acceptance of this work in Nature Communications.

Reviewer #2 (Remarks to the Author):

The current version after revision could be accepted as is.

Reviewer #3 (Remarks to the Author):

The authors have address the comments raised by the reviewers, and I recommend the publication of the manuscript.